# Etiology and Pathogenesis of Rheumatoid Arthritis-Interstitial Lung Disease

**DOI:** 10.3390/ijms241914509

**Published:** 2023-09-25

**Authors:** Yerin Kim, Hyung-In Yang, Kyoung-Soo Kim

**Affiliations:** 1Department of Medicine, Catholic Kwandong University College of Medicine, Gangneung 25601, Republic of Korea; elle_rossy@naver.com; 2Division of Rheumatology, Department of Internal Medicine, College of Medicine, Kyung Hee University Hospital at Gangdong, Seoul 05278, Republic of Korea; yhira@khu.ac.kr; 3East-West Bone & Joint Disease Research Institute, Kyung Hee University Hospital at Gangdong, Seoul 05278, Republic of Korea; 4Department of Clinical Pharmacology and Therapeutics, Kyung Hee University School of Medicine, Seoul 02447, Republic of Korea

**Keywords:** rheumatoid arthritis, interstitial lung disease, pathogenesis

## Abstract

Interstitial lung disease (ILD) is one of the most serious extra-articular complications of rheumatoid arthritis (RA), which increases the mortality of RA. Because the pathogenesis of RA-ILD remains poorly understood, appropriate therapeutic strategies and biomarkers have not yet been identified. Thus, the goal of this review was to summarize and analyze the reported data on the etiology and pathogenesis of RA-ILD. The incidence of RA-ILD increases with age, and is also generally higher in men than in women and in patients with specific genetic variations and ethnicity. Lifestyle factors associated with an increased risk of RA-ILD include smoking and exposure to pollutants. The presence of an anti-cyclic citrullinated peptide antibody, high RA disease activity, and rheumatoid factor positivity also increase the risk of RA-ILD. We also explored the roles of biological processes (e.g., fibroblast–myofibroblast transition, epithelial–mesenchymal transition, and immunological processes), signaling pathways (e.g., JAK/STAT and PI3K/Akt), and the histopathology of RA involved in RA-ILD pathogenesis based on published preclinical and clinical models of RA-ILD in animal and human studies.

## 1. Introduction

Rheumatoid arthritis (RA) is a systemic autoimmune disease that is primarily accompanied by articular manifestations, including continuous inflammation and joint deterioration [1]. RA can also affect organs and systems beyond joints. The extra-articular manifestations of RA involve several organ systems, including the skin, eyes, heart, lungs, kidneys, nervous system, and gastrointestinal system [2]. The most common extra-articular manifestation is pulmonary involvement, which affects up to 60% of patients with RA [3,4]. RA-related pulmonary diseases include interstitial lung disease (ILD), bronchiectasis, and pleural diseases, which are generally detected using chest computed tomography (CT) imaging [5].

ILD, which is a pulmonary manifestation of RA, was first described by Ellman and Ball in 1948 [6]. ILD occurs in approximately 10% of patients [7,8]. The prevalence of ILD was about 10–19% among RA patients [8,9,10]. However, the exact prevalence is not well established and varies depending on the methods of measurements [10]. The noticeable clinical symptoms of RA-ILD include exertional dyspnea, persistent dry cough, general fatigue, and weakness [11]. Because these symptoms are easily neglected, RA patients should be monitored steadily for RA-associated pulmonary symptoms. In medical practice, pulmonary function testing and high-resolution computed tomography are performed to diagnose RA-associated pulmonary diseases [12]. The major histopathological patterns observed in patients with RA-ILD include nonspecific interstitial pneumonia (NSIP) and usual interstitial pneumonia (UIP) patterns [13,14,15]. Other less commonly observed patterns include organizing pneumonia (OP) and obliterative bronchiolitis [16,17]. The representative images of UIP, NSIP, and OP are present in Figure 1. The three most seen histological appearances in RA-ILD are well represented in the review article written by Wijsenbeek et al. [18]. ILD causes lung dysfunction and is the leading cause of death in patients with RA after cardiovascular disease [19,20,21,22]. According to several studies, RA-ILD is a life-shortening disease. Raimundo et al. found that 35.9% of patients died five years after being first diagnosed with RA-ILD, and the median survival was 7.8 years in the United States [23]. Up to 7% of RA-related deaths are associated with RA-ILD [8]. Complications of ILD greatly affect the prognosis of RA because RA-ILD is generally progressive and only restricted therapeutic options are available [24]. Due to a shortage of definitive prognostic indicators or gold standard therapeutic strategies for RA-ILD, high mortality rates have been observed in these patients [25,26]. Therefore, annual screening for ILD is highly recommended for patients with RA [27].

Management of RA-ILD is challenging. Although several therapeutic agents have been proposed, large randomized controlled trials have not yet been conducted, and appropriate guidelines for clinical practice are not available [24]. Antifibrotic medications, including nintedanib, have been shown to inhibit the progression of RA-ILD; however, more clinical studies are required to verify these findings [28,29]. Moreover, antirheumatic medications such as methotrexate are not recommended for the treatment of RA-ILD because they can cause pulmonary toxicity [30,31]. Therefore, it is necessary to develop drugs that can be used to treat both arthritis and pulmonary fibrosis.

The development of novel treatments for RA-ILD requires a better understanding of the molecular mechanisms underlying its pathogenesis. Therefore, an understanding of the risk factors and pathogenesis of RA-ILD is required. Thus, this review’s focus provides a thorough overview of several risk factors and mechanisms that contribute to the development of ILD in patients with RA, although there are already several review articles on RA-ILD [32,33].

## 2. Risk Factors Associated with RA-ILD

Although the exact mechanism by which RA-ILD occurs remains unknown, genetic factors such as age, sex, race, smoking, pollutants, and autoantibodies, especially rheumatoid factors (RF) and anti-cyclic citrullinated peptides (CCP) antibody (ACPA), have been proposed as risk factors.

### 2.1. Genetic Factors

In an exome sequencing study of patients with RA-ILD, associations between RA-ILD and mutations in familial pulmonary fibrosis-linked genes (*TERT*, *RTEL1*, *PARN*, or *SFTPC*) were identified. These mutations are more frequently observed in patients with RA-ILD than in controls [34]. In a large RA cohort study, the *MUC5B* promoter variant was associated with higher odds of RA-ILD by more than two-fold. As this variant is less common in African-American patients, its existence in this population causes >four-fold increased odds of RA-ILD [35]. This association has been reported in previous studies. Evidence that the *MUC5B* promoter variant *rs35705950* can serve as a strong risk factor for RA-ILD, especially in patients with UIP patterns, has been demonstrated [36]. A large observational cohort study demonstrated that the risk of ILD was 16.8% for *MUC5B* carriers and 6.1% for *MUC5B* non-carriers among patients with RA. This difference between risks began to appear at 65 years of age, with an increased risk among men [37]. Hayashi et al. demonstrated that the *rs6578890* single nucleotide polymorphism in the PPFIA-binding protein 2 (*PPFIBP2*) gene was significantly associated with the occurrence of RA-ILD in a genome-wide association study [38]. Le Guen et al. reported the case of a 37-year-old female patient with a heterozygous *NKX2.1* mutation associated with RA-ILD and a histological pattern of lymphocytic interstitial pneumonia [39]. From a genome-wide association study meta-analysis, variants of *RPA3-UMAD1* were identified as a novel risk factor for RA-ILD in the Japanese population [40]. *rs2609255G* is a risk allele for UIP and ILD in Japanese patients with RA. *FAM13A rs2609255* was significantly associated with UIP in male and older patients with RA [41]. Similarly, in patients with RA in northern Sweden, *FAM13A* was associated with ILD, as analyzed using GWAS [42].

### 2.2. Age

Physiological changes occur in the lungs with aging and induce deterioration of lung function and increased susceptibility to diseases. Age-related declines, such as immunosenescence and inflammaging can reduce the regenerative capacity of the lung and trigger lung fibrosis [43,44]. Older age serves as a critical risk factor for RA because: (1) patients with RA live longer with better medical management and (2) more people are being diagnosed with RA at an older age. Because elderly patients with RA generally have other age-related diseases, they require different treatments than younger patients with RA [45]. An inception cohort of patients with RA with a 20-year follow-up reported that the incidence of RA-ILD was associated with older age [46]. Lai et al. found that the average age of patients with RA-ILD was higher than that of patients with RA without ILD, implying that RA-ILD is associated with advanced age [47]. RA-ILD is most commonly diagnosed at age 50–59. Age has been found to be an independent risk factor for the development of ILD in previous cohort studies [48,49].

### 2.3. Sex

In multivariate analysis, male sex was identified as a variable significantly associated with RA-ILD [50,51]. Male sex has also been identified as an independent predictor of co-occurring RA and lung diseases such as ILD, bronchiectasis, and nodules [52]. Men have a more than two-fold higher risk of RA-ILD compared to women [53]. Male sex was also significantly associated with unfavorable outcomes in patients with RA-ILD, with an HR of 2.52 [54]. Gao et al. found that the age-standardized mortality rates ratio of RA-ILD to RA was greater in men than in women. This implied that the proportion of RA patients who died from ILD was higher in men than in women [55]. In addition, it was shown that the proportion of female RA-ILD patients was higher in the non-progressor group (85%) than in the progressor group (68%) [56].

In a retrospective study of patients with RA-ILD, although most patients with RA were female, the ratio of males in the patients with RA-ILD was significantly higher than in patients with RA without ILD [47]. Most patients with RA are women; thus, these findings can be considered paradoxical, but are not explained by potential confounding factors, such as cigarette smoking [33].

However, Olson et al. reported that RA-ILD-associated mortality rates decreased in men but increased in women between 1988 and 2004 [8]. It was observed that RA incidence was predominant in females as opposed to males, with a male-to-female ratio of 1:3.1 [57]. Female sex was also an independent risk factor for the development of RA-ILD in a multi-ethnic Malaysian cohort study [58].

In a large cohort study, the early onset of menarche and irregular menstrual cycles were significantly associated with an increased risk of RA, whereas breastfeeding for >12 months was protective against RA risk [59]. In a population-based cohort study conducted in Norway, the total time of lactation was associated with reduced RA mortality, which was close to a dose–response relationship [60]. These results could partly explain why the RA risk differs according to sex.

### 2.4. Race

Jeganathan et al. demonstrated that RA- and RA-ILD-associated mortality rates vary substantially depending on the patient’s race. In general, RA-ILD-associated mortality was highest in Hispanic individuals (26% higher than white individuals), followed by white individuals, and lowest in black individuals (27% lower than white individuals) in the United States between 2005 and 2018 [61]. Native American individuals are more susceptible to RA than Caucasian individuals [62]. In a multi-ethnic Malaysian cohort study, Indian patients with RA demonstrated a significantly increased risk of developing RA-ILD [58]. Moreover, drug-induced interstitial pneumonia was significantly more frequent in the Japanese population than in people of other ethnicities, and acute exacerbation of IPF also occurred more frequently in the Japanese population than in other ethnicities [63,64].

### 2.5. Smoking

Previous reports have identified smoking as a predictive factor for RA-ILD [51,65,66,67,68]. Smoking ≥ 30 packs/year significantly increased the risk of RA-ILD, while a lower level of smoking did not increase this risk [69]. From a retrospective cohort study of tobacco smoking in ILD patients, it was found that tobacco smokers with ILD showed clinical symptoms closely associated with PF and emphysema, shorter time to lung function decline, and reduced survival [70]. The trend for association between smoking and ILD was also observed in a large US database [71]. The largest study on RA-ILD in the UK demonstrated that smoking was highly associated with ILD in males, which could explain the higher incidence of RA-ILD in men than in women [53]. RA-ILD is associated with smoking among patients with *HLA–DRB1* shared epitope (SE) [50]. The chemical constituents of cigarette smoke can also trigger an immune response, and the resultant serum autoantibodies against citrullinated proteins accumulate in the lungs, causing inflammation and injury to epithelial cells, finally resulting in ILD [11]. However, contradictory results have been reported. Mori et al. showed that the significant association between smoking history and ILD acquired from univariate analysis in patients with RA with lung problems (ILD or airway disease) was not verified by multinomial logistic regression analysis [72].

### 2.6. Pollutants

A recent study revealed that higher levels of ambient air pollutants, including PM2.5, PM10, sulfur dioxide (SO_2_), and nitrogen dioxide (NO_2_), increase the risk of hospitalization in patients with RA-ILD [73]. Silica exposure in the workplace environment is also suggested to be associated with an increased risk of developing RA and related lung manifestations [74,75,76]. Air pollutant exposure is positively associated with the development of several pulmonary diseases, especially ILDs [77]. Few studies have shown that O3 exposure causes T cell reaction dysregulation and that quenching effects have an inverse association with the incidence of ILDs [78,79,80]. By analyzing newly onset patients with RA-ILD from the United States health care insurance database (MarketScan) (2011–2018), it was identified that exposure to elements of PM2.5, especially ammonium, mineral dust, and black carbon, caused a higher ILD risk than the other PM2.5 [81].

### 2.7. Anti-Cyclic Citrullinated Peptide Antibody

Anti-cyclic citrullinated peptide antibody (ACPA) is a group of auto-antibodies that bind to citrullinated epitopes specific to RA, occurring years before the clinically evident disease appears [82]. ACPA is commercially available and typically used in clinical applications [83]. Previous studies have shown that the specificity and sensitivity of ACPA detection for the diagnosis of RA are 96–99% and 47–88%, respectively, depending on the features of the RA population [84,85]. ACPA titers are significantly increased in patients with RA-ILD compared to those in patients with RA without ILD [21,86,87], suggesting that abnormal citrullination may contribute to the development of lung fibrosis in patients with RA. Klester et al. found that the ACPA titer was the most strongly associated risk factor for RA-ILD in a univariate analysis [51]. In addition, patients with high-positive ACPA titers (>15 U/mL) showed higher risk of prevalent ILD (OR, 1.91; 95% confidence interval [CI], 1.04–3.49) than that in ACPA-negative patients [88].

A systematic review and meta-analysis showed that the existence of ACPA was significantly associated with RA-ILD incidence, and autoantibody levels was significantly increased in patients with RA-ILD compared to patients with RA alone. Nevertheless, the authors declared that their findings may be limited in application because of the heterogeneity of the included studies [83]. In a previous study performed in the UK with 230 patients with RA-ILD, ACPA was identified as the strongest predictor of RA-ILD [53]. Giles et al. found that the levels of all specific ACPA were higher in patients with radiographic evidence of ILD than in those without RA-ILD. In contrast, the levels of antibodies against non-citrullinated proteins were not higher in patients with ILD [89]. A meta-analysis reported that serum ACPA positivity is strongly associated with the risk of RA-related pulmonary diseases, especially RA-related ILD and idiopathic pulmonary fibrosis (IPF) [90]. In several cohorts, ACPA were positively associated with IPF and ectopic lymphoid aggregates in the lungs [91]. Higher ACPA titers have also been demonstrated as predictors of ILD in a multi-ethnic Malaysian cohort study [58]. Moreover, in Korean patients with RA, the ACPA titer ratio was higher in the airway disease and ILD groups than that in the control group [92].

In addition, a significantly higher positive association of ACPA has been reported in patients with RA-ILD than in patients with RA alone [93]. Citrullinated proteins accumulate in bronchoalveolar lavage fluid of patients with RA-ILD and IPF [94]. ACPA was highly associated with RA-ILD in both sexes [53]. Restrepo et al. found that variables including ACPA and RF were significantly associated with RA-ILD in multivariable analysis. These associations became stronger when the concentrations of ACPA and RF increased [50]. Citrullination not only exists in the synovial tissue of patients with RA but is also present in the affected extra-articular tissue. Samara et al. identified that the citrullination pathway was upregulated in bronchoalveolar lavage cells of patients with RA-ILD. In particular, the mRNA and protein expression levels of PAD4, which catalyzes citrullination, are elevated in RA-ILD. Moreover, the levels of citrullinated proteins were positively correlated with the levels of PAD4 and ACPAs in RA-ILD compared to the control group [94]. Moreover, the citrullinating enzyme PAD2 is upregulated in the lungs and associated fibroblasts of patients with RA-ILD when compared to the control group. PAD2 serves as a pro-fibrotic mediator in fibroblasts of patients with RA-ILD, and its suppression reduces fibroblast–myofibroblast transition (FMT) and extracellular matrix formation [95].

The anti-mutated citrullinated vimentin antibody titer in patients with RA-ILD was significantly higher than in patients with RA without ILD. However, there was no significant difference in ACPA levels between the two groups [96].

However, controversy over the association of anti-cyclic citrullinated peptide and RF with the development of RA-ILD still exists [52,97,98]. In a previous study, no significant differences were observed between the levels of ACPA and RA-associated pulmonary involvement, including ILD and bronchiolitis [97]. This discrepancy can be attributed to the obscure distinction between ILD and idiopathic pulmonary fibrosis, owing to their unclear definitions and the relatively small number of studies included in the systematic review [90].

### 2.8. Disease Activity of RA

It is well established that low disease activity or remission of disease activity is correlated with better therapeutic results for RA [99,100]. Disease activity has been identified as a risk factor for RA-ILD using the Disease Activity Score in 28 joints (DAS28) [101] or Clinical Disease Activity Index (CDAI) [102] as estimation methods.

Zhuo et al. showed that patients with RA-ILD had a greater disease burden than those with RA without ILD. RA disease activity may worsen after ILD diagnosis compared to the preclinical ILD status and patients with RA alone [103]. Sparks et al. found that moderate and high disease activity was associated with a two-fold elevation in RA-ILD risk compared to remission/low disease activity. Reducing systemic inflammation by treating RA symptoms may prevent the manifestation of RA-ILD [101]. Moderate or high joint disease activity was considered as an independent factor associated with RA-ILD, with an OR of 3.03 [104]. Patients with RA-ILD with high disease activity during follow-up showed decreased survival compared to those with moderate and low disease activity [105]. Moreover, in a nested case-control study that matched incident RA-ILD to RA without ILD, moderate to high disease activity was associated with an increased risk of RA-ILD, implying that the regulation of disease activity may reduce risk [69]. Another case-control study demonstrated that a CDAI score > 28 was associated with the incidence of RA-ILD [102]. These results suggest that controlling systemic inflammation is important for preventing RA-ILD development. Thus, regular measurement of clinical disease activity may help identify patients with RA at a higher risk for complications [103]. Moreover, among middle-aged female RA-ILD patients, the DAS28 level was significantly increased in the RA-ILD group compared to the RA group [106].

### 2.9. RF

The RF titers showed a monotonic relationship with the prevalence of ILD. Only patients with RF concentrations >90 IU/mL are at increased risk of incident RA-ILD [88]. Smoking and high RF titers (hazard ratio [HR], 1.09; 95% CI, 1.001–1.11) have been identified as risk factors for RA-ILD development [51]. Moreover, RF titers are closely related to joint injuries and RA airway lesions [107,108,109].

RF positivity has been identified as a risk factor for RA-ILD [68]. RF positivity was also identified as an independent predictor of RA-ILD development in a multiethnic Malaysian cohort study [58]. In a retrospective cohort study of patients with RA-ILD, high-titer RF seropositivity was significantly correlated with mediastinal lymph node enlargement, CT honeycombing, and reduced transplant-free survival [110].

It was demonstrated that RF-immunoglobulin (Ig)A levels >200 RU/mL (HR, 3.17, *p* = 0.012) were regarded as a predicting factor for poor prognosis of RA-ILD from the multivariate analysis [54]. In addition, the positive ratio of RF-IgA was higher in patients with RA-ILD than that in patients with RA without ILD, whereas the positive ratios of RF-IgG, RF-IgM, anti-AKA antibodies, anti-APF antibodies, and ACPA were not significantly different between the two groups [96].

### 2.10. Combination of Factors

A combination of age, sex, smoking status, RF, and ACPA was highly associated with the incidence of RA-ILD (areas under the curve are 0.88 for subjects enrolled in the Brigham and Women’s Hospital Rheumatoid Arthritis Sequential Study [111] and 0.89 for American College of Rheumatology cohorts) [87]. Patients with a combination of RF/ACPA seropositivity had a higher possibility of ILD than seronegative patients (OR, 2.90; 95% CI, 1.24–6.78). Nonetheless, combined RF/ACPA seropositivity was not associated with an increased risk of incident ILD [88].

## 3. Pathogenesis of RA-ILD

Although several risk factors associated with the development of RA-ILD have been identified, the pathophysiological connections between these factors and the pulmonary changes remain unclear.

The pathogenesis of RA-ILD is primarily related to aberrant tissue reactions in the alveolar wall and lung parenchyma [32]. The first step in the pathogenesis of RA-ILD is the deterioration of airway and alveolar epithelial cells [12]. Continuous damage to epithelial cells causes the activation of immune cells including neutrophils, dendritic cells, and macrophages, leading to the excessive accumulation of extracellular matrix (ECM) components in lung tissues [12].

Few studies have identified the mechanism of RA-ILD, which is mainly due to the limitations in establishing an animal model. As the construction of an animal model of RA representing the pathology of both articular and pulmonary manifestations is challenging, experimental studies using these models have been restricted [112,113]. Distinct from bleomycin (BLM)- or collagen-induced arthritis (CIA)-induced pulmonary fibrosis animal models, the murine model established by the combination of CIA and BLM showed significant interstitial fibrosis, involving dilatation of air gaps and thickening of interstitial walls. However, the CIA + BLM animal model is characterized by combined features of arthritis and pulmonary fibrosis and can be utilized to better study the progression of RA-ILD [114].

### 3.1. FMT

The activation of fibroblasts to become myofibroblasts and escape the quiescent state is defined as FMT and serves as a key step in fibrotic pathogenesis [115]. FMT is generally divided into two stages. After the fibroblast is activated to become a proto-myofibroblast phenotype, it enters the second stage to complete cell phenotype transition [116,117]. Myofibroblasts are derived from the excessive activation and aberrant differentiation of fibroblasts by FMT and are known to be responsible for fibrosis [118,119]. Lung epithelial cellular senescence orchestrates lung injury by stimulating FMT [120,121].

Continuous transforming growth factor (TGF)-β signaling is the major underlying mechanism of FMT and fibrosis [122]. Dysregulation of TGF-β1 and TGF-β2 signaling has been associated with the development of fibrotic diseases, including IPF [123] and systemic sclerosis-ILD [122]. TGF-β1-induced FMT causes aberrant hyperplasia of collagen in lung interstitial tissues, which results in a decline in lung function [124,125]. Following stimulation with TGF-β1 on human lung fibroblasts-1, myofibroblast transition was increased compared to the control group, as determined by a transwell assay. Additionally, staining of mouse embryonic fibroblast cells showed the elevated productions of collagen, fibronectin, and α-smooth muscle actin compared to the control group [126]. Moreover, in HLFs, TGF-β induced the upregulation of dedicator of cytokinesis 2 (DOCK2) expression at both transcriptional and post-translational levels. Dedicator of cytokinesis 2 upregulation promotes FMT and plays a role in the development of pulmonary fibrosis [127].

In murine lung fibroblasts, signal transducers and activators of transcription (STAT)3 contributed to the lung fibrosis by stimulating interleukin (IL)-6- and TGF-β1-mediated myofibroblast differentiation [128]. Furthermore, inhibition of STAT3 in fibroblasts showed reductions in TGFβ1-induced FMT and alleviated skin fibrosis in mouse models of systemic sclerosis [129]. Milara et al. also found that the dual inhibition of p-STAT3 and p-JAK2 in lung fibroblasts of patients with IPF could alleviate TGF-β1 and IL-6/IL-13-induced FMT [130]. In a BLM-induced PF mouse model, NF-κB p65 subunit was phosphorylated and translocated from cytoplasm into nucleus. Moreover, it was also found that the lung FMT was inhibited by the NF-κB phosphorylation and its nuclear translocation [131]. Wu et al. demonstrated that pirfenidone, a therapeutic agent for IPF, suppresses FMT in lung fibroblasts from patients with RA-ILD by downregulating activating transcription factor 3 [132]. It was also found that downregulated expression of cyclooxygenase-2 and prostaglandin E2 in human lung myofibroblasts is associated with FMT and epithelial–mesenchymal transition (EMT) and may have key roles in IPF development [125].

### 3.2. EMT

EMT is a cellular process in which epithelial cells acquire mesenchymal features while losing the epithelial features [133]. EMT has significant roles in the development and progression of several pulmonary diseases [134,135,136]. Tubular epithelial cells can be transformed into fibroblasts via EMT in adult kidneys [137]. EMT is involved in the accumulation of myofibroblasts and the subsequent deposition of the ECM, which are associated with ILD progression [138].

From a recent study using a RA-ILD mice model established by CIA combined with BLM-induced pulmonary fibrosis, EMT was upregulated through the TGF-β-SMAD2/3 signaling pathway in lung tissues [139]. In D1CC × D1BC transgenic mice of the RA-ILD model, the expression of Pad4 and citrullinated peptides in the lungs was upregulated, which led to an increase of EMT in epithelial cell and fibrosis [140]. In addition, M1 macrophages promote myofibroblast apoptosis and degrade the ECM via matrix metalloproteinase (MMP) activation. In contrast, M2 macrophages activate fibroblasts through the secretion of TGF-β1 and platelet-derived growth factor and secrete tissue inhibitors (TIMPs) and inhibit ECM degradation [141]. An imbalance in Th1/Th2 cells has an impact on pulmonary fibrosis. Specifically, Th1 and Th22 cells inhibit fibrosis, whereas Th2, Th9, Th17, and Tregs worsen fibrosis [141].

MMPs, which play key roles in EMT, are proteases involved in ECM degradation and are cell surface receptors [142]. Serum MMP-3 protein levels are correlated with RA disease activity [143]. MMP-3 promotes fibrogenesis by EMT activation [144]. Thus, it has been well described that MMP-3 is involved in the development of RA and lung fibrosis [145]. It has also been demonstrated that MMP-7 can be used as a biomarker of RA-ILD [146]. Another study revealed that MMP-7 is associated with RA-ILD, and risk factors (age, sex, smoking, RF, and ACPA) are known to be related to RA [87]. In a cohort of Korean patients with RA-ILD, serum MMP-7 levels were negatively correlated with the diffusing capacity for carbon monoxide and forced vital capacity measured using the pulmonary function test. Additionally, MMP-7 levels correlated with the semi-quantitative grade of CT. Thus, the authors suggested that MMP-7 can be used to measure the functional and anatomical status of lungs in patients with RA-ILD [147].

### 3.3. Immunological Pathways for Production of Different Cytokines

Chronic and continuous damage to the airway mucosa and distal lung tissue during the preclinical stages of RA can trigger innate immune and inflammatory reactions [148]. Furthermore, the constant immune stimulation and inflammation induced by RA can accelerate abnormal fibroproliferative responses [149].

As shown in Figure 2, several cytokines are involved in RA-ILD pathogenesis. From the cytokine profile analysis, MCP-1/CCL2 and SDF-1α were associated with RA-ILD, and IL-18 levels were associated with RA-ILD and more prominent progressed lung disease [150]. Tumor necrosis factor (TNF)-α is a major proinflammatory cytokine associated with the pathogenesis of interstitial lung involvement. TNF-α triggers fibroblast proliferation and the secretions of PDGF-β, TGF-β, cytokines, and chemokines [21]. TNF-α inhibitors are often used to treat RA [151]. Nonetheless, an increasing number of studies have suggested that TNF-α inhibitors are involved in ILD pathogenesis and may cause pulmonary toxicity [152]. Zhang et al. found that IL-23 serves as an initiator of fibrogenesis in RA-ILD by stimulating EMT in transitioning alveolar epithelial type I cells. IL-23 induces cells to acquire invasive properties, deposit ECM, and resist apoptosis, leading to the production of fibroblast foci in fibrotic ILD, particularly in RA [153]. Microarray profiling of primary adult pulmonary fibroblasts cultured from patients with scleroderma-associated ILD and IPF showed that IL-11 was upregulated [154].

Serum levels of IL-11 are significantly upregulated in patients with RA, which is associated with the development of ILD and disease activity, suggesting that IL-11 may be involved in the pathogenesis of RA and/or RA-ILD. It was also shown that serum IL-11 levels were positively correlated with DAS28 in patients with RA [155].

It was also reported that the histone deacetylase HDAC3 triggered the development of RA-ILD fibrosis by upregulating miR-19a-3p-mediated IL17RA in an RA-ILD mouse model induced by zymosan [156]. From a RA-ILD mice model developed by CIA with BLM-induced pulmonary fibrosis, expressions of TNF-α and IL-6 in the ankle joints of mice were upregulated compared to the control group [139]. Yang et al. observed the significantly increased levels of TNF-α, IL-6, and IL-1β and decreased levels of IL-10 in serum and lung tissues of an RA-ILD rat model [157].

Through proteomic profiling, 234 proteins were differentially expressed between the RA-ILD and RA-without-ILD groups. Gene set enrichment analysis proteins affecting these gene sets in RA-ILD included several proteins involved in lung fibrosis: cytokines (CCL18 and IL-17), chemokines (CXCL12 and CCL5), FGF family members (FGF4 and FGF7), and S-type lectin galectin-3 (LGALS3) [158]. It was determined that the serum level of sCXCL16 was significantly increased in RA-ILD patients, and CXCL16/CXCR6 could contribute to the severity of pulmonary fibrosis [159]. It was demonstrated that the expressions of CX3CL1 and CX3CR1 were up-regulated in the lung tissues of BLM-IP mouse, and the inhibition of CX3CL1 by treating anti-CX3CL1 mAb decreased the population of M1 macrophages and increased surface CD3 expressions on T cells of BALF [160]. The level of CXCL13, a B cell chemoattractant, correlates with the degree of inducible bronchial-associated lymphoid tissue in RA-ILD. Nonetheless, the exact mechanism underlying these chemokines involved in the development of RA-ILD has not been clarified [21]. Sendo et al. recently found CD11b^+^Gr^−^1^dim^ tolerogenic dendritic cell-like cells, which are distinctive suppressive myeloid cells that differentiate from monocytic myeloid-derived suppressor cells in SKG mice with ILD [161]. The CIA mouse model also represented the increased aggregation of CD11b^+^ interstitial macrophages in the subpleural inflammation area [162]. In addition, the authors demonstrated that tofacitinib triggered the expansion of MDSCs in the lungs and inhibited ILD progression in SKG mice [163].

Xu et al. showed that the serum level of soluble programmed death molecule-1, a common immunosuppressive member on the surface of T cells, in patients with RA-ILD were significantly elevated compared to that in patients with RA without ILD and healthy controls [164]. Other studies have also reported that soluble programmed death molecule-1 level are upregulated in patients with RA and are correlated with disease activity [165,166].

It was shown that anti-citrullinated HSP90β induced significantly increased interferon (IFN)-γ production in patients with RA-ILD compared to patients with RA without ILD [167]. Previous studies have also reported that anti-citHSP90 antibody positivity can identify RA-ILD with high specificity (>90%) [168]. Sibinska et al. revealed that HSP90 has a direct effect on the TGF-β1 signaling pathway and that inhibition of HSP90 decreases lung fibrogenesis and fibrosis progress in mice model [169].

### 3.4. Oxidative Stress

Autoimmune diseases are also associated with oxidative stress [170]. The levels of oxidative damage markers, including malondialdehyde and thiobarbituric acid reactive substances in patients with RA, was higher than those in healthy individuals [171,172,173]. Mitochondrial DNA mutations, consecutive respiratory chain dysfunction, and the resulting ROS accumulation have also been shown to be involved in the pathogenesis of ILD [174]. Terasaki et al. showed that oxidative stress was induced in the fibrotic lung lesions of a RA-ILD model in DICC mice, which was similar to the findings in patients with RA-ILD. In a mouse model of RA-ILD, oxidative stress was induced by increased levels of serum lipid peroxide and 8-hydroxydeoxyguanosine (8-OhdG)-positive cells. Chronic and inflammatory interstitial pneumonia was observed with increased levels of TNF-α, IL-6, and TGF-β in the lungs [175]. Wang et al. found that the serum nuclear factor erythroid 2-related factor 2 (Nrf2) levels in patients with RA were significantly associated with ILD, and that the Nrf2-related antioxidant pathway was involved in the development of oxidative stress-mediated RA-ILD [176].

### 3.5. Autophagy

Bao et al. found that in RA-ILD, the level of autophagy was increased in the early stages, whereas it was suppressed in the later stages in fibrotic lungs using a CIA + BLM RA-ILD mouse model [114]. Vasarmidi et al. revealed that the expression of BECLIN1, which helps in the initiation of autophagosome formation, was significantly upregulated in bronchoalveolar lavage fluid cells from patients with RA-ILD compared to patients with IPF. Nonetheless, other major molecules involved in the autophagy pathway were comparable between patients with IPF and RA-ILD [177].

Several studies have shown that autophagy is dysregulated in IPF lungs, highlighting the need for further research [178]. Suppression of autophagy caused by reduced expression of forkhead transcription factor O subfamily member 3a (FoxO3a) increases the viability of IPF fibroblasts. Thus, the dysregulation of autophagy induced by FoxO3a suppression may mediate IPF development [179].

Autophagy was inhibited by upregulation of the TGF-β1-Smad3/ERK/P38 signaling pathway, which lead to an increase of activated myoblasts and collagen accumulation in the BLM-induced pulmonary fibrosis mice model [180,181]. It was also demonstrated that the levels of autophagy-related markers such as LC3I/II and beclin-1 were downregulated in the lungs of rats with BLM-induced fibrosis [130]. It has also been demonstrated that suppression of miR-33 in macrophages alleviates lung fibrosis by augmenting autophagy in in vivo and ex vivo models of IPF [182].

### 3.6. Janus Kinase/Signal Transducers and Activators of Transcription Pathway

Previous studies have demonstrated that inhibition of the Janus kinase (JAK)/signal transducers and activators of transcription (STAT) pathway was effective in protecting against RA [183]. RA-ILD has also been associated with the JAK/STAT pathway [184]. JAK/STAT signaling is upregulated in fibrotic disorders of several organs, including the lungs, and it has been reported that tofacitinib, a JAK inhibitor, is effective in a murine model of ILD [163]. In a post hoc analysis of data from 21 tofacitinib clinical trials, the incidence rate of ILD was 0.18 per 100 patient-years after tofacitinib treatment [185]. Using a CIA mouse model, it has been reported that the administration of tofacitinib, a potent and selective inhibitor of JAK, significantly attenuates the development of arthritis [186,187,188].

The JAK/STAT pathway is involved in lung fibrosis [189]. Protein expression of JAK, STAT, and the receptor activator of nuclear factors κB ligand (RANKL) were upregulated in the lung tissues of a RA-ILD rat model, suggesting that the JAK/STAT/RANKL signaling pathway is involved in RA-ILD pathogenesis [157]. The RA-ILD mouse model showed upregulated protein expression of JAK2, STAT3, and phosphorylated STAT3 in the ankle joints compared to the control group [139]. Clinical evidence for the inhibitory roles of JAK inhibition in RA-ILD has been continuously reported [190]. Numerous clinical and experimental studies have demonstrated the protective effects of JAK inhibitors against connective tissue disease-ILD [191]. JAK2 is phosphorylated in the lung tissue of patients [130]. It was also found that JAK2 served as a regulator of fibrogenesis in RA-UIP and IPF as well as TGF-β signaling in both normal and lung fibrosis [192]. It has been suggested that inhibition of JAK1 and JAK2 could relieve pain in patients with RA. The JAK-signal transducer and activation of the JAK/STAT pathway are involved in the regulation of numerous inflammatory cytokines [193]. The protein expressions of the Jak2/Stat3 signaling pathway-related markers were significantly increased in the lung tissues of the CIA group compared to the control group [194]. In a CIA mouse model, angiogenesis was increased by activation of the JAK/STAT signaling pathway [195].

The JAK/STAT pathway is activated by several cytokines, including IL-4, IL-13, IL-6, IL-11, and IL-31, which are involved in the pathogenesis of ILD [184]. Because of their key roles in immune reactions and connections with several cytokine receptors [196], the inhibition of JAKs is regarded as a potential therapeutic strategy for autoimmune diseases [197]. IL-11 has been shown to participate in lung fibrosis through STAT3 activation in a BLM-induced fibrosis mouse model [198,199]. Moreover, IL-11 induces STAT3 phosphorylation. However, the effect of IL-11-induced STAT activation on transcriptional alterations and its relevance to fibrogenesis has not been clearly identified [200]. JAK inhibitors can regulate the activation, proliferation, and differentiation of B cells, indicating a key role of B cells in the development of RA from early disease onset [196]. Stattic, a STAT3 inhibitor, mitigated the lung fibrosis in a mouse model of RA-ILD. It also alleviated TGF-β1-induced inflammation, myofibroblast activation, oxidative stress, and hyper-proliferation by regulating the JAK1/STAT3 pathway [201].

### 3.7. Phosphoinositide-3-Kinase/Protein Kinase B Pathway

The upregulated phosphoinositide-3-kinase (PI3K)/protein kinase B (Akt) signaling pathway facilitates the repair and development of ILD [202]. PI3Kγ specifically has effects on the antigen-induced arthritis (AIA) during the early stage of inflammation, when innate immune cells play crucial roles in pathogenesis. As shown in Figure 3, PI3Kγ affects the migration and activation of macrophages, and macrophage and neutrophil infiltration into the knee joints is also involved. Upregulation of the citrullinating enzyme PAD2 in the lung tissues of patients with RA-ILD is controlled by syndecan-2 via regulation of the PI3K/Akt/Sp1 pathway, which is dependent on CD148 [95].

Global mRNAs and microRNAs were profiled from the lung tissues of patients with several types of ILD, and differentially expressed genes were analyzed. Several apoptosis-associated genes, including PI3K, receptor-interacting serine-threonine kinase 1 (RIP1), and bcl2-associated X protein (BAX), are downregulated, and this anti-apoptosis signaling may facilitate the survival of myofibroblasts in the lungs of patients with ILD [203]. Integrated metabolomics and network pharmacological analysis combined with PCR showed that the Ras and PI3K-Akt signaling pathways were up-regulated in RA-ILD [204]. In addition, serum IL-11 levels are substantially increased in patients with RA, which was associated with the development of ILD [155]. After IL-11 binds to its receptor, it activates the JAK/STAT3, ERK, and PI3K/AKT/mTORC1 signaling pathways to exert its effects [205,206]. It was also determined that CXCL16/CXCR6 contributed to pulmonary fibrosis in RA-ILD patients by up-regulating proliferation and collagen accumulation of human pulmonary fibroblasts (MRC-5 cells) mediated by the PI3K/AKT/FOXO3a pathway [159].

## 4. Histopathological Type of RA

Although the overall RA-related mortality is decreasing, RA-ILD with a UIP pattern shows an increased risk of disease progression and mortality [113]. UIP is generally characterized by a straight edge, anterior upper lobe, and exuberant honeycombing [207].

Chen et al. suggested that the UIP pattern is a significant risk factor for RA-ILD progression [54]. RA-ILD with a UIP histological pattern is associated with an increased risk of disease progression and mortality [17]. A prospective study demonstrated that two out of three patients with RA showing a prominent ground-glass pattern on high-resolution computed tomography scans experienced spontaneous regression of the disease compared to the group with a reticular pattern [208]. Park et al. found that patients with RA-ILD with UIP had a worse survival rate than patients with NSIP [209].

Patients with RA-ILD with UIP have a worse prognosis than those with OP [210,211]. Suhara et al. identified several proteins that may be involved in the development of differences between UIP and OP patterns in RA-ILD pathogenesis. The concentrations of gelsolin and Ig κ chain C regions were significantly upregulated in the UIP pattern compared to the OP pattern. In contrast, the levels of α-1 antitrypsin, CRP, haptoglobin β, and surfactant protein A were significantly increased in the OP pattern compared to those in the UIP pattern. In addition, the gelsolin levels were significantly higher in the UIP group than those in the OP pattern [212]. These findings suggest that distinguishing the UIP pattern from the other patterns in patients with RA-ILD may lead to better therapeutic outcomes.

## 5. Conclusions

While ILD affects a substantial proportion of patients with RA and is associated with a decreased survival rate, reliable therapeutic agents and screening biomarkers are not currently available because of the lack of understanding of the mechanism underlying the development and progression of RA-ILD. Thus, in this review, we attempted to analyze the previously reported literature and evidence of the etiology and pathogenesis of RA-ILD, as shown in Figure 4. Based on previously reported data, older age, male sex, smoking, and exposure to specific pollutants (e.g., PM2.5, PM10, SO_2_, and NO_2_) are associated with an increased risk of RA-ILD. Specific genetic variations are independent risk factors for RA-ILD. Regarding the underlying molecular mechanisms of RA-ILD, the activation of biological processes (FMT, EMT, and immunological pathways) and signaling pathways (JAK/STAT and PI3K/Akt) could be deleterious and may contribute to the development of RA-ILD. Upregulated autophagy and oxidative stress are also involved in the pathogenesis of RA-ILD. Future research may provide more concrete evidence regarding the underlying mechanisms of RA-ILD. Further experimental studies are needed to investigate the pathogenesis of RA-ILD. Moreover, patients with RA should undergo high-resolution computed tomography and pulmonary function tests to determine whether they have certain pulmonary problems at an early stage and to provide proper treatment.

## Figures and Tables

**Figure 1 ijms-24-14509-f001:**
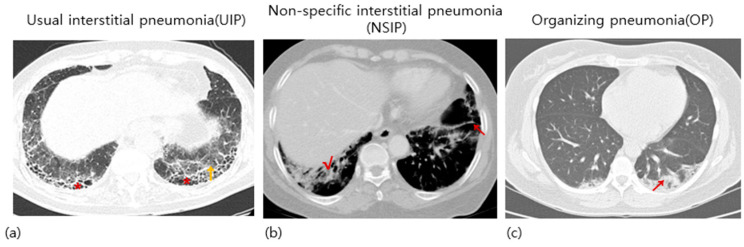
The most common three patterns in RA-ILD. The representative images of chest high-resolution CT with (**a**) UIP, (**b**) NSIP, and (**c**) OP patterns are shown. (**a**) Bilateral subpleural and basal reticular opacities (**↑**) with honeycomb (*****) are typical of a UIP pattern. (**b**) Subpleural patchy parenchymal opacities (**√**) with underlying fibrosis and traction (**↖**) in both lungs. (**c**) Subpleural consolidations (**↗**).

**Figure 2 ijms-24-14509-f002:**
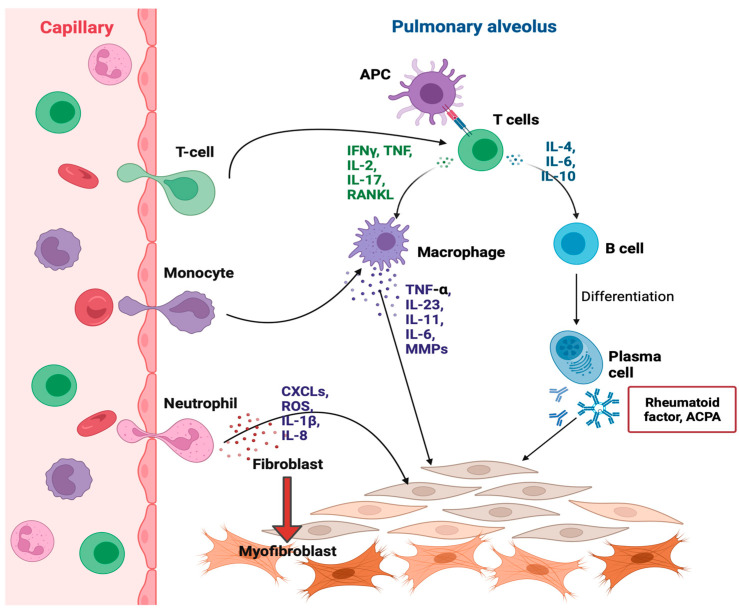
Schematic diagram of the immunological pathway involved in the pathogenesis of RA-ILD. Activated macrophages, T cells, and neutrophils secrete several pro-inflammatory cytokines (narrow black arrows) to promote lung inflammation and fibrogenesis by up-regulating FMT (bold red arrow).

**Figure 3 ijms-24-14509-f003:**
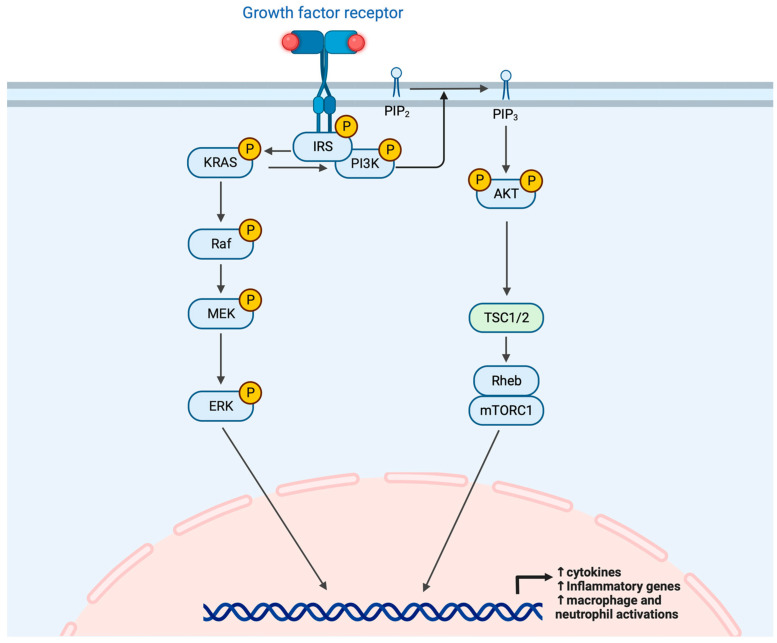
Schematic diagram of the PI3K/Akt pathways involved in RA-ILD pathogenesis. As illustrated, the phosphorylation of PI3K activates Akt through PIP2 to PIP3 conversion. PI3K/Akt/mTORC1 and ERK contributes to the up-regulations of target genes and proteins, which promote the inflammation by regulating macrophage, neutrophil, and pro-inflammatory cytokines.

**Figure 4 ijms-24-14509-f004:**
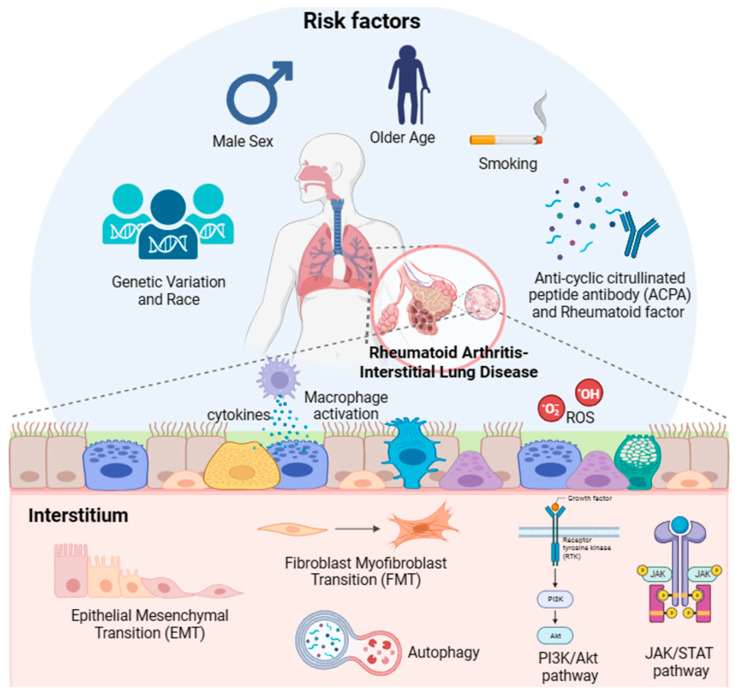
Schematic diagram of risk factors that induce rheumatoid arthritis-interstitial lung disease (RA-ILD) and proposed pathogenesis. Although the exact pathophysiological mechanisms in which RA-ILD is developed remains unknown, several risk factors and plausible mechanisms have been proposed. Risk factors associated with RA-ILD include genetic variations, male sex, older age, race, smoking, exposure to pollutants, and anti-cyclic citrullinated peptide antibody (ACPA). Reported plausible pathophysiological mechanisms involve biological processes (e.g., fibroblast–myofibroblast transition [FMT], epithelial–mesenchymal transition [EMT], and immunological processes), signaling pathways (e.g., JAK/STAT and PI3K/Akt), and RA histopathology.

## Data Availability

No new data were created or analyzed in this study. Data sharing is not applicable to this article.

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
