# Peer review of "Etiology and Pathogenesis of Rheumatoid Arthritis-Interstitial Lung Disease"

_ijms, 2023, doi:10.3390/ijms241914509_

Round 1

Reviewer 1 Report

This review provides a detailed introduction to the etiology and pathogenesis of Rheumatoid Arthritis-Interstitial Lung Disease. Although there are many unknowns about the RA-ILD, this article is a useful source of information with the evidence to date. However, the Histopathological type of RA section on page 12 would be better understood by the reader if representative histological figures were added. It would be desirable to revise this section if possible.

Author Response

We would like to appreciate the reviewers for their every constructive advice and critiques, which have assisted in improving our review article, “Etiology and Pathogenesis of Rheumatoid Arthritis-Interstitial Lung Disease”. We have now addressed all of the reviewer’s comments and questions, as well as our corresponding point-by-point response to reviewer comments in the following paragraphs. The edited section is indicated in red.

Reviewer 1’s comments 

This review provides a detailed introduction to the etiology and pathogenesis of Rheumatoid Arthritis-Interstitial Lung Disease. Although there are many unknowns about the RA-ILD, this article is a useful source of information with the evidence to date. However, the Histopathological type of RA section on page 12 would be better understood by the reader if representative histological figures were added. It would be desirable to revise this section if possible.

Response: Thank you for the valuable comment. We newly added the representative images of UIP, NSIP, and OP (Figure 1, page 2) in the section of Introduction. However, we could not get the pictures of the histopathology of the respective type of RA-ILD due to lack of the patients’ samples. The three most seen histological appearance in ILD is shown in Lancet 2022;400:769-86. It is added to reference list.

Reviewer 2 Report

The novelity of manuscript is underquestion. English language has good quality. The manuscriot needs some figures in some sections.

1. Please explain what is the difference between your manuscript and two manuscript below:

1. Rheumatoid arthritis-interstitial lung disease: manifestations and current concepts in pathogenesis and management

Suha Kadura, Ganesh Raghu

European Respiratory Review 30 (160), 2021"

2. Rheumatoid arthritis-associated interstitial lung disease: Current update on prevalence, risk factors, and pharmacologic treatment

Sicong Huang, MD, MS, Vanessa L. Kronzer, MD, MSCI, Paul F. Dellaripa, MD, Kevin D. Deane, MD, PhD, Marcy B. Bolster, MD, Vivek Nagaraja, MD,Dinesh Khanna, MD, MSc, Tracy J. Doyle, MD, MPH, and Jeffrey A. Sparks, MD, MMSc

Current treatment options in rheumatology 6, 337-353, 2020

2. The authors should insert two figure for two section of manuscript including:

" 3.3. Immunological pathways for production of different cytokines" in page 8-9

" 3.7. Phosphoinositide-3-kinase/protein kinase B pathway" in page 11-12

3. Please check and adjust the "Reference list" based on the regulations of reference list of journal. (Titles, doi, the name of journal and ... )

Author Response

Dear Editor-in Chief,

We would like to appreciate the reviewers for their every constructive advice and critiques, which have assisted in improving our review article, “Etiology and Pathogenesis of Rheumatoid Arthritis-Interstitial Lung Disease”. We have now addressed all of the reviewer’s comments and questions, as well as our corresponding point-by-point response to reviewer comments in the following paragraphs. The edited section is indicated in red.

 Reviewer 2’s comments

  1. Please explain what is the difference between your manuscript and two manuscript below:
  2. Rheumatoid arthritis-interstitial lung disease: manifestations and current concepts in pathogenesis and management

Suha Kadura, Ganesh Raghu

European Respiratory Review 30 (160), 2021"

  1. Rheumatoid arthritis-associated interstitial lung disease: Current update on prevalence, risk factors, and pharmacologic treatment

Sicong Huang, MD, MS, Vanessa L. Kronzer, MD, MSCI, Paul F. Dellaripa, MD, Kevin D. Deane, MD, PhD, Marcy B. Bolster, MD, Vivek Nagaraja, MD,Dinesh Khanna, MD, MSc, Tracy J. Doyle, MD, MPH, and Jeffrey A. Sparks, MD, MMSc

Current treatment options in rheumatology 6, 337-353, 2020

Response: Thank you for the critical comment. Those review articles both dealt with the etiology of RA-ILD, similar to our article. However, those articles did not only focus on the etiologic aspect, but also included treatment approaches of RA-ILD. Therefore, both articles could not present a comprehensive analysis of the current data in relation to the pathogenesis of RA-ILD.
    To be specific, ‘Rheumatoid arthritis-interstitial lung disease: manifestations and current concepts in pathogenesis and management’ article focused on the pathogenesis of RA-ILD in the context of immunological mechanism only. On the contrary, our review article attempted to provide a thorough summary of pathogenesis of RA-ILD. Thus our article summarized the pathogenesis of RA-ILD by dividing into several subsections including FMT, EMT, immunological pathway, oxidative stress, autophagy, JAK/STAT, and PI3K/Akt pathway.
   ‘Rheumatoid arthritis-associated interstitial lung disease: Current update on prevalence, risk factors, and pharmacologic treatment’ review article did not cover the pathogenetic aspects of RA-ILD at all. It only summarized the risk factors of RA-ILD.
   Our review article aimed to provide complete and in-depth understandings of pathogenesis of RA-ILD, especially in focus of the molecular mechanisms performed by experimental procedures. Moreover, our article tried best to update the latest reports in pathogenesis and etiology of RA-ILD. However, according to the comments, the above two references is added to reference list for readers. We clarify the difference between them at the end of Introduction.

  1. The authors should insert two figure for two section of manuscript including:

" 3.3. Immunological pathways for production of different cytokines" in page 8-9

" 3.7. Phosphoinositide-3-kinase/protein kinase B pathway" in page 11-12

Response: Thank you for the comment. We newly added two figures for 3.3 section (Figure 2) and 3.7 section (Figure 3). 

  1. Please check and adjust the "Reference list" based on the regulations of reference list of journal. (Titles, doi, the name of journal and ... )

Response: Thank you for the comment. We revised the "Reference list" based on the regulations of reference list of the journal. 

Reviewer 3 Report

Authors are requested to do the suggested changes.

Minor English corrections are needed. 

Few sentences are difficult to read.

Author Response

Dear Editor-in Chief,

We would like to appreciate the reviewers for their every constructive advice and critiques, which have assisted in improving our review article, “Etiology and Pathogenesis of Rheumatoid Arthritis-Interstitial Lung Disease”. We have now addressed all of the reviewer’s comments and questions, as well as our corresponding point-by-point response to reviewer comments in the following paragraphs. The edited section is indicated in red.

Reviewer 3

Major Comments

  1. What about involvement of Environmental factors in RA-ILD?

Response: Thank you for the comment. For environmental factors, smoking and several pollutants have been recognized to be the risk factors of RA-ILD. We also summarized the reported results and suggested mechanisms of smoking and pollutants on increasing RA-ILD risk (2.5 and 2.6 sections, page 4). Do we need to describe other factors? If you suggest, we will edit it. 

  1. Authors can outline “Prevalence and clinical features of RA-associated lung diseases”.

Response: Thank you for the comment. As the reviewer suggested, we newly added the paragraph about the prevalence and clinical features of RA-ILD. “The prevalence of ILD was about 10%–19% among RA patients [8-10]. The exact prevalence, however, is not well established and varies depending on the methods of measurements [10]. The noticeable clinical symptoms of RA-ILD include exertional dyspnea, persistent dry cough, general fatigue, and weakness [11]. Since these symptoms are easily neglected, RA patients should be monitored steadily for the RA-associated pulmonary symptoms. In medical practice, pulmonary function testing and high-resolution computed tomography, are performed to diagnose RA-associated pulmonary diseases [12].” was added in the Introduction, page 1-2.

  1. Review is too lengthy and does not offer much new insights. Most of the information is

already available.

Response: Thank you for the comment. Our review article tried to provide thorough and in-depth understandings of pathogenesis of RA-ILD, especially in context of the molecular mechanisms proven by experiments. As the reviewer pointed out, we deleted several citations that were reported before early 2010s. Additionally, we newly cited more recent references which were published after 2020.

  1. Authors need to include more figures and tables.

Response: Thank you for the comment. We newly added three more figures, such as CT images of the UIP, NSIP, and OP (Figure 1) and schematic diagrams for 3.3 section (Figure 2) and 3.7-3.8 sections (Figure 3).

  1. More than 2/3 of references were before 2020. Authors need to use more recently published articles and reviews.

Response: Thank you for the comment. We newly cited more recent references which were published after 2020.

Minor Comments

  1. Abbreviation should only be used if the intended is present more than thrice.

Response: Thank you for the comment. We revised the manuscript to use abbreviations when the word was present more than thrice. 

Round 2

Reviewer 2 Report

 No more comments.